# Isoflavone Metabolism by Lactic Acid Bacteria and Its Application in the Development of Fermented Soy Food with Beneficial Effects on Human Health

**DOI:** 10.3390/foods12061293

**Published:** 2023-03-18

**Authors:** Susana Langa, Ángela Peirotén, José Antonio Curiel, Ana Ruiz de la Bastida, José María Landete

**Affiliations:** Departamento de Tecnología de Alimentos, Instituto Nacional de Investigación y Tecnología Agraria y Alimentaria (INIA-CSIC), Carretera de La Coruña Km 7.5, 28040 Madrid, Spain

**Keywords:** soy, isoflavones, genistein, equol, LAB, bioavailable, aging, health

## Abstract

Isoflavones are phenolic compounds (considered as phytoestrogens) with estrogenic and antioxidant function, which are highly beneficial for human health, especially in the aged population. However, isoflavones in foods are not bioavailable and, therefore, have low biological activity. Additionally, their transformation into bioactive compounds by microorganisms is necessary to obtain bioavailable isoflavones with beneficial effects on human health. Many lactic acid bacteria (LAB) can transform the methylated and glycosylated forms of isoflavones naturally present in foods into more bioavailable aglycones, such as daidzein, genistein and glycitein. In addition, certain LAB strains are capable of transforming isoflavone aglycones into compounds with a greater biological activity, such as dihydrodaidzein (DHD), *O*-desmethylangolensin (*O*-DMA), dihydrogenistein (DHG) and 6-hydroxy-*O*-desmethylangolensin (6-OH-*O*-DMA). Moreover, *Lactococcus garviae* 20-92 is able to produce equol. Another strategy in the bioconversion of isoflavones is the heterologous expression of genes from *Slackia isoflavoniconvertens* DSM22006, which have allowed the production of DHD, DHG, equol and 5-hydroxy-equol in high concentrations by engineered LAB strains. Accordingly, the consequences of isoflavone metabolism by LAB and its application in the development of foods enriched in bioactive isoflavones, as well as health benefits attributed to their consumption, will be addressed in this work.

## 1. Introduction

Isoflavones are a group of phenolic compounds considered phytoestrogens due to their similar structure to estrogens [1]. Isoflavones are present in many members of the family Fabaceae and their main dietary source is from soy and soy products [2]. These compounds are associated with beneficial effects on human health, such as a decreased risk for cardiovascular disease, improvement of menopausal quality of life, a reduction in the risk of some kinds of cancer and the improvement of neurodegenerative diseases [3,4]. Several of these beneficial effects acquire special relevance during aging and, given the worldwide increase of the elderly population [5], confers an important role to soy products as functional foods.

However, isoflavones are found in nature mainly in their glycosylated and methylated forms, which are not absorbed directly by the intestine. These forms must be metabolized by the intestinal microbiota to become bioavailable and, subsequently, have a higher biological activity [6,7]. Therefore, the benefits provided to human health by isoflavones are highly influenced by the host intestinal microbiota [8,9,10,11]. The presence of specific bacteria or bacteria communities is key to the formation of bioactive isoflavones, such as genistein, equol and *O*-desmethylangolensin (*O*-DMA), which will have different effects on human health compared to their precursors [12,13].

There are discrepancies in the beneficial health effects that the consumption of isoflavone-containing food has since the benefits are dependent on the intestinal microbiota, and there is great variability in the intestinal microbiota present in individuals [14,15]. Therefore, it would be relevant to consume foods containing bioactive isoflavones, a bacterial strain or a group of bacterial strains capable of producing them, or both, such as fermented soy products [16,17,18]. For these reasons, there is a great interest in the metabolism of isoflavones by LAB and bifidobacteria [14]. The selection of LAB and bifidobacteria strains, able to produce bioactive isoflavones, could be utilized in food for the production of fermented foods with high bioactive isoflavones concentrations as most of the LAB species hold the qualified presumption of safety (QPS) and are generally recognized as safe (GRAS) to consume [19]. In this review, we will discuss how LAB and bifidobacteria transform and modify the isoflavones found in food, the effects of metabolism and their potential benefits for human health.

## 2. Transformation of Glycosylated and Methylated Isoflavones into Their Aglycones by LAB

In soy and other legumes rich in isoflavones, isoflavones exist mainly in their *O*-glycosylated, *C*-glycosylated or methylated forms [12]. Glycosylated and methylated isoflavones are less estrogenic and antioxidant than their respective aglycones, and they are poorly absorbed by the human intestine because of their higher hydrophilicity and molecular weights [20]. Their bioavailability requires their conversion into aglycones by bacteria, such as LAB and bifidobacteria, by the gut microbiota or in the fermented food [7,8,21]. Several pharmacokinetic studies have shown a faster and higher absorption of isoflavones, as well as extended levels in plasma, after ingestion of aglycones compared with glycosides [22,23,24], although others have shown contradictory results and a consensus is yet to be stablished [7,25].

Daidzin, genistin and glycitin are the most abundant isoflavones found in nature and are *O*-glycosylated isoflavones. LAB strains are among the bacteria that produce β-glycosidases able to produce the aglycones daidzein, genistein and glycitein from daidzin, genistin, and glycitin, respectively [26,27] (Figure 1). Studies on the fermentation of soy food by LAB have revealed that a high percentage of the studied LAB strains were able to deglycosylate the isoflavone glycosides present in soy food leading to their corresponding aglycones, with the most widespread trait being the deglycosylation of genistin [21]. However, the efficiency in the transformation of glycoside isoflavones into their corresponding aglycones varies greatly between species but mainly between strains [26,28,29,30,31]. Like this, the fermentation of soy beverages with different LAB strains resulted in overall daidzein and genistein production ratios ranging from 14% to 60%, with *Limosilactobacillus mucosae* INIA P508 being able to transform the initial content of 780 μM glycosides and 8 μM aglycones in the beverage by into nearly 460 μM of aglycones [32].

Kudzu is a food rich in a *C*-glycoside of daidzein named puerarin [33]. The bioavailability and biological activity of puerarin is dependent on its *C*-deglycosylation (Figure 1). The transformation of puerarin to daidzein was scarcely found among the LAB strains tested by different research groups [27,34]. Breakage of the *C*-glycoside bond seems to be more difficult in this glycoside and these data are in agreement with the less frequent detection of this metabolism by gut microbiota [35]. Kim, Lee and Han [34] found that two LAB strains, *Lactococcus* sp. MRG-IFC-1 and *Enterococcus* sp. MRG-IFC-2, were capable of hydrolyzing the *C*-glycosidic bond of puerarin, whereas Gaya, Peirotén and Landete [23] identified two *Enterococcus faecalis* strains, INIA P90 and INIA P1, which were able to hydrolyze puerarin into daidzein.

Biochanin A and formononetin are the methoxylated derivatives of genistein and daidzein, respectively, found in plants and herbs, such as red clover [36]. These methylated isoflavones need to be hydrolyzed to the aglycones daidzein and genistein by the appropriate *O*-demethylases [37] (Figure 1). Thus, biochanin A was metabolized to genistein in soy beverage by *O*-demethylation carried out by *Lacticaseibacillus rhamnosus*, *Lactiplantibacillus plantarum* and *Limosilactobacillus fermentum* strains, whereas the *Lactococcus lactis* and *Lacticaseibacillus paracasei* strains studied were not able to transform biochanin A [38]. On the other hand, none of the lactobacilli, lactococci or bifidobacteria strains studied by Curiel and Landete [38] were able to demethylate formononetin in the soy beverage.

Therefore, LAB are promising candidates for the fermentation of soy food due to the ability of some of their strains to metabolize isoflavones with high efficiency, reaching high levels of isoflavone aglycones in the functional fermented soy beverages [15,18,21,29]. Recent studies have shown that the refrigeration of the product after fermentation, as well as the thermal treatment and subsequent storage at room temperature after such fermentation, seem suitable conservation procedures that did not affect the concentration of bioactive isoflavones present in the fermented soy beverages [18].

## 3. Metabolism of Isoflavone Aglycones by LAB

Daidzein and genistein aglycones can be transformed into DHD and DHG, respectively, by hydrogenation reactions [12,39] (Figure 2 and Figure 3). LAB strains were capable of transforming daidzein into DHD with low efficiency in a fermented soy beverage, whereas DHG could not be detected in the same beverages [32]. Recently, *Ligilactobacillus acidipiscis* HAU-FR7 was shown to be capable of efficiently transforming daidzein and genistein into DHD and DHG, respectively, in a soy beverage [40].

LAB are also capable of producing *O*-DMA and 6-hydroxy-*O*-DMA (6-OH-*O*-DMA) (Figure 2 and Figure 3). *Enterococcus hirae* AUH-HM195 [41] and *Enterococcus faecium* INIA P553 [42] were the first LAB strains identified as being capable of producing *O*-DMA and 6-OH-*O*-DMA (Table 1). More recently, the production of these compounds by different LAB species in fermented soy beverages has been described [32], observing that a high percentage of the tested strains were capable of producing *O*-DMA and 6-OH-*O*-DMA in fermented soy beverages. The strains that produced the highest concentrations of *O*-DMA and 6-OH-*O*-DMA in soy beverages were *L. mucosae* INIA P508, which produced 21.32 ± 1.21 µM of *O*-DMA and *Limosilactobacillus reuteri* INIA P572, which produced 24.22 ± 7.65 µM of 6-OH-*O*-DMA [32]. Recently, we observed that *O*-DMA and 6-OH-*O*-DMA were metabolized mainly from daidzein and genistein by LAB strains through fission of the *C*-ring [43], although *O*-DMA and 6-OH-*O*-DMA could also be produced from DHD and DHG with lower efficiency. Thus, the increase in daidzein and genistein concentration in soy beverages produces an increase in the concentration of *O*-DMA and 6-OH-*O*-DMA by the LAB producers of these bioactive isoflavones.

DHD and DHG must be transformed into tetrahydrodaidzein (THD) and tetrahydrogenistein (THG) for their subsequent transformation into equol and 5-hydroxy-equol (5-OH-equol), respectively. These isoflavones are considered the most interesting due to theirs beneficial human health effects [13]. Besides *Lactococcus garviae* 20-92 [45], THD production has only been shown in some LAB strains, such as *L. paracasei* INIA P461, and always with low efficiency [32] (Table 1), while THG production and 5-hydroxy-equol have not been observed in any LAB to date, with the exception of genetically modified LAB strains [49].

Regarding the production of equol, the prevalence of equol producers (individuals capable of metabolizing daidzein into equol by their gut microbiota) has been reported to be between 20% and 35% among Western adults [50]. Equol production from daidzein by a single bacterium has been related with the family *Coriobacteriaceae*, more specifically with the genus *Eggerthella* spp., *Slackia* spp. and *Adlercreutzia* spp. [13]. In addition, the genes involved in equol production were also identified and sequenced in *Lactococcus garvieae* 20-92 [51], showing a genetic organization similar to that described in *Coriobacteriaceae*, suggesting the horizontal transference of these genes from a member of this family to *L. garvieae* [13].

Although the production of equol by LAB strains can be considered strange, equol production has been observed not only in *L. garvieae* 20-92, but also by *Lactobacillus intestinalis* JCM 7548, *L. paracasei* CS2, *Lactilactobacillus sakei* CS3 and *Pediococcus pentosaceus* CS1 from *Pueraria lobata* extract [46] and *L. intestinalis* KTCT13676BP from daidzein and chungkookjang [47] (Table 1). On the other hand, LAB are able to produce equol or help to increase the equol production by their association with other phylogenetically distant strains. The mixture of *Lactobacillus* sp. Niu-O16 and *Eggerthella* sp. Julong 732 strains, which was known to transform daidzein into DHD and DHD to equol, resulted in a significant increase in the equol production when compared to the production of *Eggerthella* sp. Julong 732 alone [52]. Moreover, the fermentation of soy beverages with the consortium of *L. fermentum* DPPMA114, *L. plantarum* DPPMA24W and DPPMASL33 and *L. rhamnosus* DPPMAAZ1 resulted in equol production [48].

## 4. Genetic Engineering as a Strategy for the Production of Bioactive Isoflavones by LAB

As above mentioned, LAB have a high capacity to produce bioactive isoflavones. However, not all LAB strains are capable of producing these bioactive isoflavones, and only a few bacteria strains have been identified as being capable of producing equol, while LAB produce THD, DHD and DHG with low efficiency. Even LAB strains producing some bioactive isoflavones, such as THG or 5-OH-equol, have not yet been identified. Recently, different studies have resorted to the genetic engineering of LAB for the production of bioactive isoflavones, such as daidzein, genistein, DHD, DHG, THD, THG, *O*-DMA, 6-OH-*O*-DMA or even equol and 5-OH-equol, by strains with biotechnological and/or probiotic interest for the development of fermented foods enriched in these bioactive isoflavones [49,53].

### 4.1. Heterologous Expression of Degycosylases and Demethylases from GRAS Bacteria

Regarding the metabolism of isoflavones and the production of bioactive isoflavones, the main characteristic of LAB is the transformation of glycoside isoflavones into their aglycones, which has already been discussed. Nevertheless, since not all LAB show a high efficiency for the β-deglycosylation of daidzin, genistin and glycitin, the gene of a β-deglycosidase (*gly*913) was identified in the genome of *L. mucosae* INIA P508, cloned and expressed in different LAB strains [54] with the aim of increasing the efficiency of β-deglycosylation.

LAB harboring pNZ:TuR.gly913 or pLEB590.gly913 (a food grade vector), both including *gly*913 under the promoter of elongation factor Tu of *L. reuteri* CECT 925, improved the ability to transform glycoside isoflavones into their aglycones [54]. In addition, the increase in the production of daidzein and genistein by the engineered strains produced a rise in the production of *O*-DMA and 6-OH-*O*-DMA by LAB with the ability to produce these compounds [43]. Thereby, soy beverages containing high *O*-DMA, 6-OH-*O*-DMA, daidzein and genistein concentrations would be of great value for the development of functional foods [43].

Isoflavone demethylase activity is not as frequent as deglycosylase activity in LAB, so the identification and cloning of genes involved in isoflavone demethylation was of great interest. Thus, the *O*-demethylase gene (*dmt*734) was identified from the genome of *Bifidobacterium breve* INIA P734 and it was cloned in pNZ:TuR.dmt734 (a vector expressing antibiotic resistance), as well as in pLEB590.dmt734 (a food grade vector). These constructions were subsequently transformed in *O*-demethylase lacking LAB strains [38], which were able to transform biochanin A into genistein with high efficiency but did not transform formononetin into daidzein, according to that which was observed in non-transformed LAB strains. This showed a specificity in the demethylation of isoflavones. These results confirmed the biotechnological interest of *O*-demethylase from *B. breve* INIA P734 for the development of fermented functional foods enriched in genistein [38].

### 4.2. Heterologous Expression of a Daidzein Reductase Gene Involved in the Production of DHD and DHG

LAB strains produce a low DHD concentration in soy beverages; however, they are unable to produce DHG in the same conditions [32]. Since both DHD and DHG are bioactive isoflavones and can promote the production of equol and 5-OH-equol in equol producing individuals, it would be of interest to produce both DHD and DHG in high concentrations. To do this, the daidzein reductase gene (*dzr*) from *S. isoflavoniconvertens* DSM22006 was cloned under a strong constitutive promoter and expressed in LAB strains, and the heterologous expression of the *dzr* gene in the different transformed LAB strains lead to a great amount of DHD and DHG in soy beverages from pure daidzein and genistein [44]. Moreover, in a colonic environment, *Lactococcus lactis* MG1363 pNZ:TuR.dzr, which expresses recombinant daidzein reductase, increased the generation of equol by equol-producing intestinal microbiota [44].

### 4.3. Soy Beverage Enriched in Equol and 5-Hydroxy-Equol

To date, equol production by LAB has been described in a small number of strains. Additionally, no LAB able to produce 5-OH-equol has been identified. Since equol and 5-OH-equol are the isoflavones of greatest interest for human health, it was of great interest to clone the genes involved in equol production. Therefore, in order to produce equol and 5-OH-equol via GRAS bacteria, the daidzein reductase gene (*dzr*), the dihydrodaidzein reductase gene (*ddr*), the tetrahydrodaidzein reductase gene (*tdr*) and the dihydrodaidzein racemase (*ifc*A) gene were amplified from the *S. isoflavoniconvertens* DSM22006 genome (Figure 4A) and cloned in a vector with a strong and constitutive promoter. The engineered LAB strains harboring these genes were able to produce equol from daidzein [53]. The next step was to produce equol and 5-OH-equol in soy beverages using these engineered LAB; however, the use of soy beverages as a base for this kind of product should take into account that its isoflavones are mainly in the form of glycosides [12]. Thus, to obtain an efficient production of equol, the glycosidase activity should be ensured in the form of another bacterial strain with high glycosidase activity. Likewise, the co-fermentation of soy beverages with the engineered *L. fermentum* INIA P584L harboring *dzr*, *ddr* and *tdr* and *Bifidobacterium pseudocatenulatum* INIA P815, a bacteria with high glycosidase activity [18], allowed the production of equol and 5-OH-equol [49] (Figure 4B). Moreover, equol and 5-OH -equol concentrations were increased during fermentation in the different soy beverages tested upon the addition of *L. fermentum* INIA P584L harboring *ifc*A [49] (Figure 4C). The high concentrations of equol and 5-OH-equol obtained in these beverages have great potential for consumer health [13].

## 5. Effects of the Isoflavones Metabolism by LAB

### 5.1. Increased Bioavailability of Isoflavones

Phenolic compounds’ bioavailability is greatly variable; therefore, the relative percentage that is excreted in urine (as a percentage of intake) ranges from 0.3% for anthocyanins to 43% for isoflavones [55]. This bioavailability can be even lower when the food polyphenols have a large molecular weight, as is the case for glycosylated and methylated isoflavones because of their high molecular weight and hydrophilicity [20,56]. Therefore, their bioavailability requires the conversion to aglycones via the action of β-glycosidases, *O*-glucosidases and/or *O*-demethylases from bacteria that colonize the small intestine for uptake into the peripheral circulation [57]. LAB by means of β-glycosidases, *O*-glucosidases and/or *O*-demethylases metabolize these isoflavones from food into smaller isoflavones, which are better absorbed in the intestine [56,58]. Then, fermentation by LAB strains enhance the absorption rate of isoflavones from soy food [26].

The intake of isoflavones has been related for years with the improvement of the lipid profile, prevention of cardiovascular disease and the alleviation of symptoms associated with menopause [1]. Thus, much of the research on the beneficial influence of isoflavones on human health have been focused on these aspects. In this sense, soy beverages fermented by *L. paracasei* NTU 101 was effective in preventing hyperlipidemia and atherosclerosis [59]. In addition, a decrease in the risk of atherosclerosis and cardiovascular diseases by the consumption of soybean products fermented by LAB strains was observed by Chen, et al. [60], and this was linked to the increase of aglycone isoflavones and their antioxidant capacities.

### 5.2. Increased Antioxidant Activity of Fermented Foods

In addition to the improvement in their bioavailability, the increased antioxidative activity of fermented soy food is attributed primarily to bioactive isoflavones, such as daidzein and genistein produced from isoflavones present in soy foods [30], although other factors, such as the production of bioactive peptides, have been also described [61]. Therefore, a good linear correlation between the antioxidant activity and the isoflavone aglycones was demonstrated [62]. Moreover, *O*-DMA and equol possess greater antioxidant properties compared with their precursor daidzein because the antioxidant activity of isoflavones increases with their transformation [63]. In this context, the antioxidant activity of fermented soy food varies depending on the LAB used in the fermentation, but it is significantly higher than that of unfermented soy food [64].

The antioxidant activity attributed to isoflavones could be either by modulating the intracellular antioxidant enzymes or scavenging the free radicals [65]. Daidzein and genistein protect membrane lipids from peroxidative damage by increasing the activity of superoxide dismutase and catalase [66]. Moreover, bioactive isoflavones protect against oxidative DNA damage exhibiting a stronger antioxidant activity [30].

### 5.3. Estrogenic/Anti-Estrogenic Effect

Isoflavones are named phytoestrogens because they bind to estrogen receptors (ERα and ERβ), which are translocated to the nucleus and bind to a DNA sequence recognized as an estrogen response element [67,68]. The binding of ERβ and ERα to the estrogen response element produces the induction of gene activation and transcription [67], which is known to have beneficial effects on human health [69,70].

However, as previously described, isoflavones found in nature, such as methylated and glycosylated isoflavones, show low estrogenic activity, and this activity increases when isoflavones are metabolized. Therefore, the estrogenic activity of *O*-DMA and equol is higher than that of daidzein [71,72] because *O*-DMA and equol are able to activate the binding of both ERβ and ERα to the estrogen response element better than diadzein [73]. Moreover, it was demonstrated that equol induces the expression of an estrogen-responsive protein gene in MCF-7 cells more effectively than daidzein [72]. LAB improves the estrogenic/anti-estrogenic activity of isoflavones found in nature through their metabolism. Thus, soy beverage fermented by LAB shows a higher estrogenic/anti-estrogenic activity [74].

In relation to menopause and the estrogenic effect of isoflavones, an equol supplement obtained by the fermentation of a soy germ by *L. garviae* 20-92 showed that menopausal women who consumed 10 mg/day of S-equol during a 12-week period reported a reduction in hot flashes and a significant improvement in sweating and irritability [75]. Dietary supplement with soy food fermented by LAB attenuated aging-induced bone loss in BALB/c mice and possibly reduced the risk of osteopenia or osteoporosis due to aging [76]. Therefore, the intake of fermented soy food with LAB could improve the beneficial effects on lipid profiles and menopause with respect to non-fermented soy beverages [77].

In relation to cancer and the estrogenic/anti-estrogenic effect of isoflavones, fermented soy beverages exhibited a higher anticancer activity due to the increased amount of genistein and daidzein, as well as free amino acids [78], and the ability of aglycones to bind to body estrogen receptors, triggering an anti-estrogenic activity [69,70].

### 5.4. Other Effects of Isoflavone Metabolism and Fermentation of Soy Foods

Bioactive isoflavones are recognized as good anti-inflammatory agents. Bioactive isoflavones, especially equol, can inhibit the inflammatory status induced by treatment with interferon-γ and lipopolysaccharide [48]. Similarly, genistein, through the inhibition of the master regulatory transcription factors GATA-3 and STAT-6, decreased airway inflammation associated with Th2-type cytokines in a murine asthmatic model [79].

Bioactive isoflavones exhibit antiproliferative properties and participate in the regulation of apoptosis, which are directly linked to their attributed protective role against certain types of cancer [80]. Moreover, daidzein induces apoptosis in hepatic cancer cells via the mitochondrial pathway [81]. Bioactive isoflavones inhibited the proliferation of carcinogenic cells in vitro in a dose-dependent manner [82]. Gercel-Taylor, et al. [83] demonstrated an inhibitory effect of genistein on ovarian cancer cell growth, and daidzein is effective in inhibiting the proliferation of lung cancer both in vitro and in vivo [84]. Additionally, the presence of equol and *O*-DMA has been associated with the inhibition of cancer cell proliferation and a chemoprotective role in the prevention of several cancers [85,86,87].

Beneficial effects of fermented soy food have also been observed in neurodegenerative diseases. Go, et al. [88] demonstrated that soybean products (Cheonggukjang) fermented by *Bacillus subtilis* and *Lactilactobacillus sakei* had a beneficial effect for neurodegenerative diseases, such as Parkinson’s disease and Alzheimer’s disease. Daidzein alleviates neuronal damage and oxidative stress [89]. Additionally, a neuroprotective effect of genistein has also been demonstrated [90,91].

Although soy-based foods are a good source of nutrition, sensory acceptability is the major limiting factor for its wide popularity. Fermentation of soy-bases foods by LAB not only improves the functional properties of the final product, but also improves its rheological and physicochemical characteristics and sensory properties [92,93].

## 6. Conclusions and Perspectives

The beneficial effects of isoflavones on human health depend greatly on their transformation into bioactive isoflavones by the intestinal microbiota. Since there is a high inter-individual variability in the microbiota composition, the effect of consuming soy food varies greatly between individuals and between studies. Therefore, not all studies have reached the same conclusions and many of them have found null results on the influence of isoflavones on human health. This is the reason why the beneficial effects of isoflavones are more evident with fermented soy foods enriched in bioactive isoflavones, which are bioavailable and show high estrogenic and antioxidant activity, as well as anti-inflammatory and antiproliferative activity.

LAB, together with bifidobacteria, which are GRAS and can be used in food, are of great importance for the development of functional foods enriched in bioactive isoflavones. LAB are characterized by having a high ability to produce bioactive isoflavones, mainly the aglycones daidzein, genistein and glycitein, in addition to transforming daidzein and genistein into *O*-DMA and 6-*O*-DMA, respectively. Therefore, soy food fermented with probiotic LAB strains capable of synthesizing bioactive isoflavones constitute an interesting and economically feasible biotechnology strategy that could be easily adapted by the food industry to develop novel functional foods tailored for aging people with the aim of improving their quality of life.

Through genetic engineering techniques, the biochemical machinery for the production of DHD, DHG, equol or 5-OH-equol could be transferred from other phylogenetically distant organisms, such as *S. isoflavoniconvertens*, to LAB strains, which are easier to grow and are considered GRAS. The production of bioactive isoflavones through the heterologous expression of genes is a strategy that allows the production of high concentrations of these bioactive isoflavones by LAB strains selected for their technological, probiotic or organoleptic properties. Engineered LAB could be used in the fermentation of soy beverages as long as they are not present in the final product (i.e., foreign DNA undetectable via PCR in the fermented food). Therefore, a special effort must be made to eliminate these engineered strains from the final product without negatively affecting the functional and organoleptic quality of the fermented food. On the other hand, the immobilization of the enzymes producing bioactive isoflavones in food grade supports could be an alternative in the production of fermented soy-based food enriched in bioactive isoflavones, such as of DHD, DHG, equol or 5-OH-equol.

The exploitation of equol-enriched fermented soy beverages would allow the use of equol in large-scale interventional trials to confirm its potentially beneficial effects. The search for strains that produce bioactive isoflavones and new genetic engineering strategies suitable for food products should be conducted in the future.

## Figures and Tables

**Figure 1 foods-12-01293-f001:**
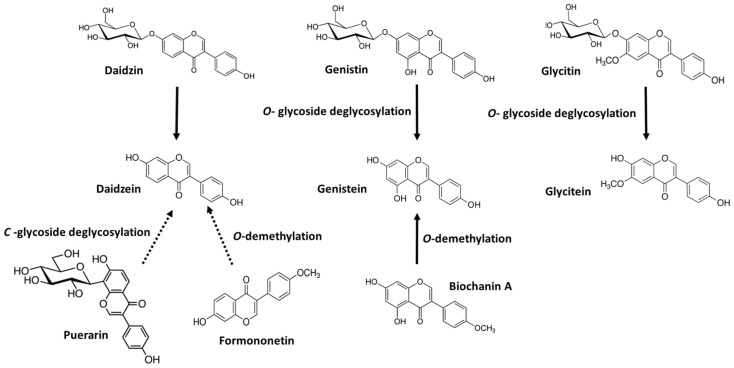
Transformation of glycosides and/or methylated isoflavones into their aglycones by LAB strains. Solid lines show frequent reactions in LAB and dashed lines show infrequent reactions.

**Figure 2 foods-12-01293-f002:**
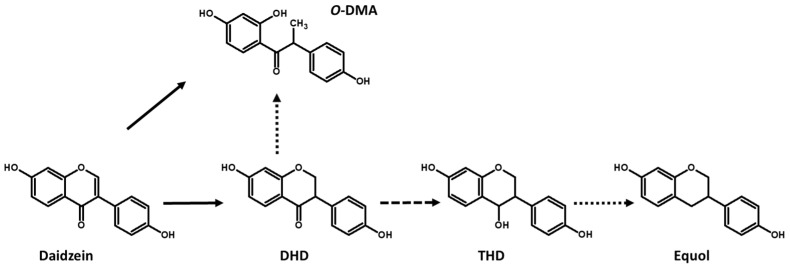
Transformation of daidzein to *O*-DMA and equol. Solid lines show frequent metabolic routes in LAB and dashed lines show infrequent reactions.

**Figure 3 foods-12-01293-f003:**
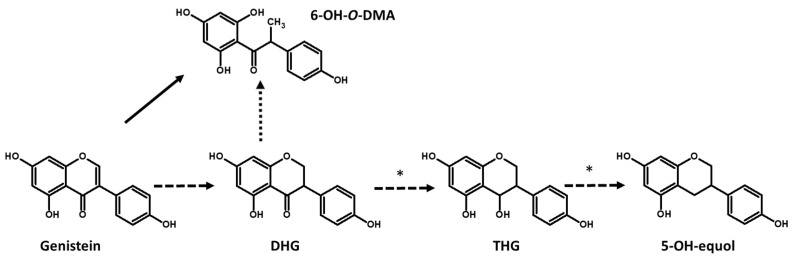
Transformation of genistein to 6-OH-*O*-DMA and 5-OH-equol. Solid lines show frequent reactions in LAB and dashed lines show infrequent reactions. * Transformation of DHG to THG and THG to 5-OH-equol by LAB in soy beverages has not been observed to date.

**Figure 4 foods-12-01293-f004:**
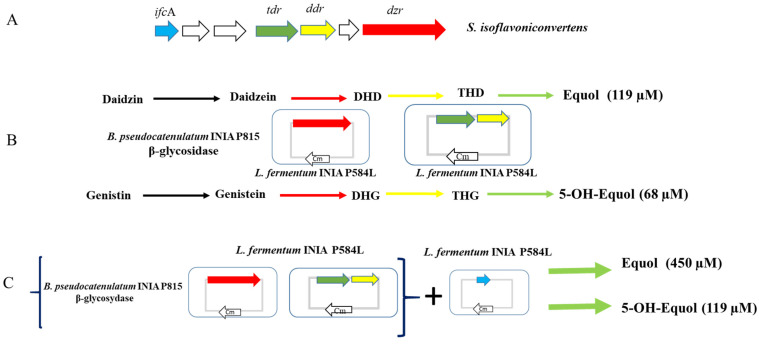
Genetic engineering as a strategy for the production of bioactive isoflavones in LAB. *Ifc*A, *tdr*, *ddr* and *dzr* genes from *S. isoflavoniconvertens* involved in equol and 5-OH-equol production (**A**). *B. pseudocatenulatum* INIA P815 transforms daidzin and genistin found in soybean into daidzein and genistein, and the heterologous expression of the *tdr*, *ddr* and *dzr* genes in *L. fermentum* INIA P584L allow the production of equol and 5-hydroxy-equol (**B**). The incorporation of *ifc*A increases the production of equol and 5-OH-equol (**C**).

**Table 1 foods-12-01293-t001:** LAB strains involved in the production of bioactive isoflavones.

Strain	Phytoestrogen Precursor	Product of Metabolism	Reference
*L. mucosae* INIA P508	Daidzin, genistin and glycitin	Daidzein, genistein and glycitein	[18]
*Enterococcus* sp. MRG-IFC-2	Puerarin	Daidzein	[34]
*Lactococcus* sp. MRG-IFC-1	Puerarin	Daidzein	[34]
*E. faecalis* INIA P90	Puerarin	Daidzein	[27]
*E. faecium* INIA P1	Puerarin	Daidzein	[27]
*L. rhamnosus* INIA P540	Biochanin A	Genistein	[38]
*L. plantarum* ESI144	Biochanin A	Genistein	[38]
*L. paracasei* INIA P272	Biochanin A	Genistein	[38]
*L. fermentum* INIA 584L	Biochanin A	Genistein	[38]
*L. acidipiscis* HAU-FR7	Daidzein and genistein	DHD, DHG	[40]
*E. hirae* AUH-HM195	Daidzein and genistein	*O*-DMA; 6-hydroxy-*O*-DMA	[41]
*E. faecium* INIA P553	Daidzein and genistein	*O*-DMA; 6-hydroxy-*O-*DMA	[42]
*L. plantarum* ESI144	Daidzein and genistein	*O*-DMA; 6-hydroxy-*O*-DMA	[44]
*L. rhamnosus* INIA P540	Daidzein and genistein	*O*-DMA; 6-hydroxy-*O*-DMA	[44]
*L. paracasei* INIA P461	Daidzein	THD	[44]
*L. garvieae* 20-92	Daidzein	Equol	[45]
*P. pentosaceus* CS1	Pueraria extract	Equol	[46]
*L. intestinalis* JCM 7548	Pueraria extract	Equol	[46]
*L. paracasei* CS2	Pueraria extract	Equol	[46]
*L. sakei* CS3	Pueraria extract	Equol	[46]
*L. intestinalis* KTCT13676BP	Daidzein/chungkookjang	Equol	[47]
*L. fermentum* DPPMA114, *L. plantarum* DPPMA24W and DPPMASL33, and *L. rhamnosus* DPPMAAZ1	Soy beverage	Equol	[48]

## Data Availability

No new data were created or analyzed in this study. Data sharing is not applicable to this article.

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
