# Peer review of "Isoflavone Metabolism by Lactic Acid Bacteria and Its Application in the Development of Fermented Soy Food with Beneficial Effects on Human Health"

_foods, 2023, doi:10.3390/foods12061293_

Round 1

Reviewer 1 Report

The review "Isoflavone metabolism by lactic acid bacteria and its application in the development of fermented soy food with beneficial effects on human health" discusses the use of LAB for converting isolflavones into their more bioactive derivatives. The review is in general well structured and well written. My two main comments are that I think the authors should be a bit more critical towards literature on health effects and also report if conflicting information is available. The second main comment is that the way the authors describe the use of GMOs does not make much sense. That must be revised.

Find more detailed comments below:

Is the field critical enough about studies on health outcome? For instance the cited systematic review by Naghshi 2022 mentions that the included papers in the review took along important confounding factors, but I miss for instance higher intake in a plant-based diet in the list. That in itself could be causing the found effects (and not necessarily the isolfavones which will be higher in people that heat more plant based food). Similarly literature is not 100% consistent in phytoestrogens only being positive - can the authors comment on this in general, when it comes to health outcomes e.g. ratio of positive vs negative effects found in literature?

Line 102: can the authors be a bit more specific what "a short time" is

line 176: Is the deglycosylation of of glycoside-isoflavones a process that happens outside the cell (e.g. are the deglycosilases secreted)? Is it known in literature why cells would make such enzymes in the first place? Same  question counts for demethylase.

Fig. 5: species names are not in italics

line 278: What do the authors mean when they mentions anti-estrogenic activity?

line 372: When the authors write about "vectors" what do they mean? Is this about a plasmid vector harbouring the gene of interest? The text would not make much sense in this case. Is it about the actual organisms that is called a vector here? That also seems odd. Please clarify.

line 384: The authors write "the elimination of DNA from GMOs should be the tasks to be carried out in the future". I wonder if the authors understand what a GMO is? One can not eliminate DNA from a GMO or any organism - it would be dead. If they mean one should eliminate the heterologoous/foreign DNA from the GMO  (this is the feeling I have the authors mean in line 372 -see comment above) then the GMO would not do what it is meant to do anymore, because the enzyme is gone. This section really needs to be checked by someone that understands how genetic engineering works.

Author Response

The review "Isoflavone metabolism by lactic acid bacteria and its application in the development of fermented soy food with beneficial effects on human health" discusses the use of LAB for converting isolflavones into their more bioactive derivatives. The review is in general well structured and well written. My two main comments are that I think the authors should be a bit more critical towards literature on health effects and also report if conflicting information is available. The second main comment is that the way the authors describe the use of GMOs does not make much sense. That must be revised.

Find more detailed comments below:

Is the field critical enough about studies on health outcome? For instance the cited systematic review by Naghshi 2022 mentions that the included papers in the review took along important confounding factors, but I miss for instance higher intake in a plant-based diet in the list. That in itself could be causing the found effects (and not necessarily the isolfavones which will be higher in people that heat more plant based food). Similarly literature is not 100% consistent in phytoestrogens only being positive - can the authors comment on this in general, when it comes to health outcomes e.g. ratio of positive vs negative effects found in literature?

Authors: Although phytoestrogens were firstly studied as endocrine disruptors in cattle, clinical studies in humans often report the absence of adverse effects (e.g. https://www.efsa.europa.eu/en/efsajournal/pub/4246). We agree with you that there exist controversial in the extent of the beneficial effects of isoflavones in health. Nevertheless, performing a review like that is not the subject of this work, that is based in the role that lactic acid bacteria fermentation can exert in the isoflavones present in soy foods.

Line 102: can the authors be a bit more specific what "a short time" is

Authors: The sentence has been reformulated: “Therefore, LAB are promising as starters for the fermentation of soy food due to the ability of some of their strains to metabolize isoflavones with high efficiency, reaching high levels of isoflavone aglycones in the functional fermented soy beverages

line 176: Is the deglycosylation of of glycoside-isoflavones a process that happens outside the cell (e.g. are the deglycosilases secreted)? Is it known in literature why cells would make such enzymes in the first place? Same  question counts for demethylase.

Authors: glucosidases are described to be mainly intracellular and associated with glucose acquisition through the catabolism of glucosides. Nevertheless, there are still many aspects of their physiological role to be clarified.

Fig. 5: species names are not in italics

Authors: corrections and additions have been made to figure 5

line 278: What do the authors mean when they mentions anti-estrogenic activity?

Authors: The anti -estrogenic term is a widely known term Phytoestrogens are plant-derived compounds that can exert various estrogenic and anti-estrogenic effects due to their affinity for the estrogen receptors. Phytoestrogen binds to estrogen receptors with a lower affinity to human estrogen. Binding of the phytoestrogen to the receptor may result in partial activation of the receptor (agonistic effect) or displacement of the endogenous estrogen molecule, thereby reducing receptor activation (antagonistic or anti-estrogenic effect). These apparently opposed effects depend on the concentration of the phytoestrogen, on the level of endogenous estrogen and on the target tissue or organ.

line 372: When the authors write about "vectors" what do they mean? Is this about a plasmid vector harbouring the gene of interest? The text would not make much sense in this case. Is it about the actual organisms that is called a vector here? That also seems odd. Please clarify.

Authors: In Molecular Biology, a vector is any particle used as a vehicle to artificially carry a foreign nucleic sequence into another cell, where it can be replicated and/or expressed. Among them, the most commonly used vectors are plasmids. We are aware that the term “vector” is used in other areas of knowledge such as Epidemiology, Mathematics, Physics and Computer Science, with very different meanings, but in Molecular Biology the term has only one meaning and is commonly used in the way we do in the manuscript.

line 384: The authors write "the elimination of DNA from GMOs should be the tasks to be carried out in the future". I wonder if the authors understand what a GMO is? One can not eliminate DNA from a GMO or any organism - it would be dead. If they mean one should eliminate the heterologoous/foreign DNA from the GMO  (this is the feeling I have the authors mean in line 372 -see comment above) then the GMO would not do what it is meant to do anymore, because the enzyme is gone. This section really needs to be checked by someone that understands how genetic engineering works.

Authors: The “elimination of DNA from GMOs” is referred to the final food product, not the GMOs themselves. That issue is addressed with more detail in the paragraph just above that one, now improved: “Engineered LAB could be used in the fermentation of soy beverages as long as there are not present in the final product, being that foreign DNA undetectable via PCR in the fermented food. Therefore, a special effort must be made to eliminate these engineered strains from the final product without negatively affect the functional and organoleptic quality of the fermented food”. Nevertheless, the sentence you mention has been improved for better understanding.

Reviewer 2 Report

Dear authors,

The review is fairly well-written and flows nicely. However, there are some minor grammatical and syntax errors that could use polishing. 

I really only have two suggestions that may improve this review:

1. I suggest including some quantitative numbers in the review. For example, mentioning some of isoflavone content found in some typical foods and how it can be improved upon using specific LAB.

and

2. It would be beneficial to provide more detail into the pharmacokinetic effects of isoflavone.

If the authors can elaborate a little more of those details I think it would benefit this manuscript more.

Author Response

Dear authors,

The review is fairly well-written and flows nicely. However, there are some minor grammatical and syntax errors that could use polishing. 

I really only have two suggestions that may improve this review:

  1. I suggest including some quantitative numbers in the review. For example, mentioning some of isoflavone content found in some typical foods and how it can be improved upon using specific LAB.

 Authors: Quantitative numbers on the production of aglycones have been added to section 2. “Transformation of glycosilated and methylated isoflavones into their aglycones by LAB” following your advice. Also quantification of equol and 5-OH- equol is now presented in figure 4.

and

  1. It would be beneficial to provide more detail into the pharmacokinetic effects of isoflavone.

  Authors: Pharmacokinetic has been studied mainly for glycosides and aglycones, information in this regard is now included in section 2 following your advice.

Reviewer 3 Report

This manuscript titled “Isoflavone metabolism by lactic acid bacteria and its application in the development of fermented soy food with beneficial effects on human health” (manuscript ID foods-2253895) presents an overview of the potential of lactic acid bacteria (LAB) (including bifidobacterial strains) in transforming/modifying the isoflavones naturally present in foods into more bioavailable structural form of compound. Moreover, the heterologous expression of genes from phylogenetically distant organisms might represent one of new ways for the bioconversion of isoflavones generated by engineered LAB strains, and thus offer more benefits to human health. The topic of research is interesting, but there are some drawbacks existing in the current version of this manuscript. A major revision needs to be done .

Specific points are listed as follows.

1) There are many abbreviations used in this manuscript, and thus a list of abbreviations should be added.

2) Line 50-51, References are necessary.

3) Line 101-103, References are necessary.

4) Line 111, what means “in low concentration”, please clarify it.

5) Line 122, and line 125, how to understand “low concentrations” as well as “a small concentration”? Not clear.

6) Line 142-144, “the genes involved in equol production have only been identified 142 and sequenced in Lactococcus garvieae 20-92 [43]. The genes involved in equol production in L. garvieae were sequenced”, what difference between them? Please check in.

7) Line 171, please indicate “in low concentrations”.

8) Line 221, please check the logic connection between this sentence “On the other hand, no LAB has been able to produce 5-hydroxy-equol” and the former expressions from line 219-221).

9) The location referenced in Figure 5 is not found in this manuscript, and where is the Figure 4 (see line 241-248)?

10) Line 238-240, References are necessary.

11) Line 245-246, could indicate “The incorporation of ifcA 245 increases the production of equol and 5-hydroxy-equol (C)” clearly in figure 5C?

12) It should be better for reading if the section 5 and section 6 would be merged and re-organized because both of them only focus on the benefits of the isoflavones to health, either generated via LAB metabolic activity or their fermented soy foods.

13) Line 305, “soyfoods” is correct?

14) The section Conclusion and perspectives of this manuscript is not concise enough. It should be further condensed, and reduced the repeated expressions appeared in its different sections.

15) Line 378-380, How to draw this conclusion “Strategies as the elimination of the DNA or the immobilization of the enzymes implicated in the bioactive isoflavone production in soy foods, will be key in placing these fermented products on the market”, because no information on this point was specifically indicated in the current version.

16) If possible, a graphic abstract could be presented for the good summary of this manuscript.

17) The references list does not indicate the documents No. 90th and 91st. Please check in them.

18) Line 406, no page number for reference 5.

Author Response

This manuscript titled “Isoflavone metabolism by lactic acid bacteria and its application in the development of fermented soy food with beneficial effects on human health” (manuscript ID foods-2253895) presents an overview of the potential of lactic acid bacteria (LAB) (including bifidobacterial strains) in transforming/modifying the isoflavones naturally present in foods into more bioavailable structural form of compound. Moreover, the heterologous expression of genes from phylogenetically distant organisms might represent one of new ways for the bioconversion of isoflavones generated by engineered LAB strains, and thus offer more benefits to human health. The topic of research is interesting, but there are some drawbacks existing in the current version of this manuscript. A major revision needs to be done .

Specific points are listed as follows.

1) There are many abbreviations used in this manuscript, and thus a list of abbreviations should be added.

Authors: A list of abbreviations is now included

2) Line 50-51, References are necessary.

Authors: reference has been added

3) Line 101-103, References are necessary.

Authors: references have been added

4) Line 111, what means “in low concentration”, please clarify it.

Authors: it refers to low efficiency, it has been changed.

5) Line 122, and line 125, how to understand “low concentrations” as well as “a small concentration”? Not clear.

Authors: references to low concentrations have been revised and corrected.

6) Line 142-144, “the genes involved in equol production have only been identified 142 and sequenced in Lactococcus garvieae 20-92 [43]. The genes involved in equol production in L. garvieae were sequenced”, what difference between them? Please check in.

Authors: That piece has been rewritten for clarity, now: “In addition, the genes involved in equol production were also identified and sequenced in Lactococcus garvieae 20-92, showing a genetic organization similar to that described in Coriobacteriaceae, suggesting the horizontal transference of these genes from a member of this family to L. garvieae”

7) Line 171, please indicate “in low concentrations”.

Authors: “in low concentration” has been replaced by “with low effiency”

8) Line 221, please check the logic connection between this sentence “On the other hand, no LAB has been able to produce 5-hydroxy-equol” and the former expressions from line 219-221).

Authors: The sentences have been revised and corrected.

9) The location referenced in Figure 5 is not found in this manuscript, and where is the Figure 4 (see line 241-248)?

Authors: The error has been corrected, former “figure 5” is figure 4 now.

10) Line 238-240, References are necessary.

Authors: references have been added

11) Line 245-246, could indicate “The incorporation of ifcA 245 increases the production of equol and 5-hydroxy-equol (C)” clearly in figure 5C?

Authors: figure has been improved following your advice

12) It should be better for reading if the section 5 and section 6 would be merged and re-organized because both of them only focus on the benefits of the isoflavones to health, either generated via LAB metabolic activity or their fermented soy foods.

Authors: former sections 5 and 6 have been now organized in a single section according to your comments.

13) Line 305, “soyfoods” is correct?

Authors: it has been replaced by “soy-based products”

14) The section Conclusion and perspectives of this manuscript is not concise enough. It should be further condensed, and reduced the repeated expressions appeared in its different sections.

Authors: the section has been modified and condensed following your advise.

15) Line 378-380, How to draw this conclusion “Strategies as the elimination of the DNA or the immobilization of the enzymes implicated in the bioactive isoflavone production in soy foods, will be key in placing these fermented products on the market”, because no information on this point was specifically indicated in the current version.

Authors: The sentence has been rewritten for clarifying, that idea is supposed to fit in the part “perspectives” of the “Conclusions and perspectives” sections.

16) If possible, a graphic abstract could be presented for the good summary of this manuscript.

Authors: a graphical abstract has been included in the new version of the manuscript.

17) The references list does not indicate the documents No. 90th and 91st. Please check in them.

Authors: references has been updated and those typos corrected.

18) Line 406, no page number for reference 5.

Authors: reference 5 is a book, doi is now included

Round 2

Reviewer 3 Report

Authors have revised their manuscript point by point, and there are no more comments for the modified version.